# Study on the effect of modified Liu-an decoction on Th1/Th2 function in Guinea pigs with cough variant asthma through the Notch signal pathway

Fangwei Xu[1], Jian Deng[1], Ningning Zhang[2], Kimberly XinTing Leow[3], Panpan Li[4], Chen Lu[5], Yuhang Chen[2], Ye Zhang[2], Liqun Wu[6]*

1 Guangzhou Women and Children's Medical Center, Guangzhou Medical University, Tianhe District, Guangzhou, China, 2 Beijing University of Chinese Medicine, Chaoyang District, Beijing, China, 3 Singapore Thong Chai Medical Institution, Thong Chai Building, Singapore, 4 Beijing Huaxin Hospital, Chaoyang District, Beijing, China, 5 Affiliated Pediatric Hospital of Fudan University, Minhang District, Shanghai, China, 6 Dongfang Hospital, Beijing University of Chinese Medicine, Fengtai District, Beijing, China

* Wulq1211@163.com

## Abstract

### Objective

To explore the effects and mechanism of Modified Liu-an Decoction (MLAD) on Th1/Th2 function and the Notch signaling pathway in guinea pigs with cough variant asthma (CVA).

### Methods

60 SPF Hartley guinea pigs were randomized into six groups (n = 10): the blank control group, model group, montelukast sodium group (MS group), as well as MLAD group of low-dose, mid-dose and high-dose. Intraperitoneal injection of ovalbumin, aluminum hydroxide sensitization, and nebulized inhalation of ovalbumin were performed in all guinea pigs of non-blank control groupto induce CVA. Cough frequency and airway resistance were recorded. The chronic inflammatory infiltration in the airways was detected by using HE and Masson staining smears. The IL-2, IL-4, IL-12, and IL-13 levels were measured by enzyme-linked immunoassay (ELISA). The expression of Delta, Jagged1, Notch1, and NICD proteins in the lung tissues were analyzed by Western blot.

### Results

In terms of airway sensitivity, coughing time and airway resistance were reduced in the MLAD and MS groups compared to the model group (P < 0.05). MLAD reduced the inflammatory infiltration in the lungs of CVA guinea pigs compared with the model group (P < 0.05).For cytokines in BALF, the contents of IL-2 and IL-12 were increased

**Data availability statement:** The data from this studyis available from the Dryad database under the registration number DOI: https://doi.org/10.5061/dryad.x95x69pxg.

**Funding:** This study was supported by [the National Natural Science Foundation of China (No. 81874487), and the Guangzhou Science and Technology Bureau Research Project, No. 2023A04J1255] in the form of a grant awarded to [WLQ] and [XFW]. The specific roles of this author are articulated in the 'author contributions' section. The two authors are the leads of these two research projects, respectively. The funders had no role in study design, data collection and analysis, decision to publish, or preparation of the manuscript.

**Competing interests:** The authors have declared that no competing interests exist.

in the MS group and each dose group of MLAD compared with the model group, and the contents of IL-4 and IL-13 were decreased compared with them.The contents of IL-12 were significantly different in each dose group of MLAD compared with the model group; the contents of IL-2 and IL-4 differed with increasing concentrations of MLAD. 4 content with increasing MLAD concentration, the more significant difference. IL-13 content decreased significantly in the MLAD low dose group compared with the model group ($P < 0.05$). In the serum cytokine model group IL-2, IL-12, IL-13 content were significantly higher than the blank group, IL-4 content was lower than the blank group, and the difference between each intervention group and the model group was not regular.

The expression levels of Jagged1, Notch, and NICD proteins in the CVA guinea pigs in the MS group and the low-dose MLAD group were decreased compared with those in the model group, and the expression of Delta1 protein was increased compared with that in the model group ($P < 0.05$).

## Conclusion

The addition of Liuan Decoction to the lungs can clear lungs and remove heat and strengthen the spleen to resolve phlegm, congestion, edema degree, reduce airway inflammation, and at the same time can reduce peri-tracheal collagen deposition, effectively improve airway remodeling, by regulating the dynamic balance of Th1/Th2. The specific downstream regulatory mechanisms of Notch signaling pathway need to be explored in further studies.

## 1. Introduction

Cough variant asthma (CVA) is a unique type of asthma with cough as the main clinical manifestation, which often occurs in the morning and at night, and triggered by exposure to cold air and irritating odors, and aggravated after exercise. CVA typically lasts for more than a month, meanwhile, recurrent attacks can adversely affect the children's quality of life. It is reported that CVA is the leading cause of chronic cough in Chinese children, accounting for 41.95% [1,2].The optimal treatment window is often missed due to the absence of wheezing, limited clinician awareness, and diagnostic experience. Approximately 30% of untreated CVA cases in children may progress to typical asthma [3–5].The etiology and mechanisms underlying CVA are complex, and antibiotic treatments are ineffective. Currently, standard treatments include glucocorticoids, bronchodilators, and leukotriene receptor antagonists. However, due to the psychiatric side effects, parents of Chinese children are very cautious about long-term oral administration of montelukast sodium, which affects compliance and overall efficacy [6,7].Traditional Chinese medicine can play an alternative and complementary role to some extent. In recent years, traditional Chinese medicine (TCM) has shown promising results in treating CVA, with better therapeutic efficacy, fewer side effects, and higher acceptance rates among patients. Modified Liu-an

Decoction (MLAD) has been effective in treating children with phlegm-heat and phlegm-dampness syndromes, and it has been found to stimulate appetite more effectively than Montelukast sodium [8,9]. MLAD is a well-established prescription for chronic cough in children, composed of Pinellia Ternata, Citri Reticulatae, Poria Cocos, Sinapis Alba, pumice, Lepidium Apetalum, Semen Trichosanthis, Rhizoma Arisaema Cum Bile, and Raphanus Sativus. Previous animal studies have demonstrated that MLAD can reduce serum levels of tumor necrosis factor-α (TNF-α) and interleukin-5 (IL-5), decrease eosinophils (EOS) in bronchoalveolar lavage fluid (BALF), and reduce airway inflammation and collagen deposition. These effects contribute to improved airway remodeling and reduced cough frequency [10–12]. Building on these findings, this study aims to analyze the relationship between CVA airway inflammation and Th1/Th2 cell balance and explore the role of cytokines and the Notch signaling pathway in the therapeutic mechanisms of MLAD in treating CVA.

## 2 Materials and methods

### 2.1 Experimental animal

60 healthy 3-week-old pure male guinea pigs, clean SPF grade and weighing 230 to 280g, were purchased from Beijing Weitong Lihua Experimental Animal Technology Co. Ltd. (License Number: SYXK (Beijing) 2019−0013). As guinea pigs are social animals and should not be raised in a single cage, thus 5 guinea pigs were placed per cage according to their size. The guinea pigs were raised under standard conditions in the clean SPF-grade animal room of Dongfang Hospital of Beijing University of Chinese Medicine, with 12h light and night cycle, temperature (23±2)℃, humidity 50%−60%. The experimental protocol was approved by the Ethics Committee for Animal Experiments of Dongfang Hospital of Beijing University of Chinese Medicine, Ethics Approval No. 201918. The guinea pigs were euthanized by releasing carbon dioxide into the box. This experiment follows national and institutional guidelines on the care and use of laboratory animals.

### 2.2 Experimental drug

The MLAD was composed of *Pinellia Ternata* 6g, *Citrus maxima* 6g, *Poria Cocos* 6g, *Prunus Armeniaca* 6g, *Sinapis Alba* 6g, *Glycyrrhiza uralensis* 3g, pumice 20g, *Lepidium Apetalum* 6g, *Semen Trichosanthis* 10g, *Rhizoma Arisaema Cum Bile* 4g, *Raphanus Sativus* 10g. The full scientific species (Latin binomial nomenclature) names of all ingredients of MLAD are shown in Table 1. The low-dose group, mid-dose group, and high-dose group were prepared into solutions with concentrations of 0.74g/mL, 1.48g/mL, and 2.22g/mL respectively. MLAD was provided by Beijing Kangrentang Pharmaceuticals and identified by the Department of Pharmacy, Dongfang Hospital, Beijing University of Traditional Chinese Medicine. Montelukast sodium (Hangzhou Merck Pharmaceutical Co., Ltd., specification: 4mg/ tablet) was made into suspension

**Table 1. The full scientific species (Latin binomial nomenclature) names of all ingredients of Modified Liu-an Decoction.**

| Chinese name | English name | Latin name |
|---|---|---|
| BAN XIA | Ternate Pinellia | *Pinellia Ternata* |
| JU HONG | -- | *Citrus maxima* |
| FU LING | Indian Bread | *Poria Cocos* |
| XING REN | Apricot Seed | *Prunus Armeniaca* |
| BAI JIE ZI | White Mustard Seed | *Sinapis Alba* |
| GAN CAO | Licorice | *Glycyrrhiza uralensis* |
| LAI FU ZI | Garden Radish Seed | *Raphanus Sativus* |
| HAI FU SHI | Pumice | Pumex |
| TING LI ZI | Pepperweed Seed Equivalent | *Lepidium Apetalum* |
| GUA LOU ZI | Snakegourd Seed | *Semen Trichosanthis* |
| DAN NAN XING | Bile Arisaema | *Rhizoma Arisaema Cum Bile* |

with normal saline and stored in a refrigerator at 4℃. The drugs were provided by the traditional Chinese Medicine Pharmacy of Dongfang Hospital of Beijing University of Chinese Medicine and Kang Ren Tang Pharmaceutical Co., Ltd. Ovalbumin (OVA) (batch number: 421C033, Solarbio Company) and Aluminum hydroxide (batch number: 20131125, Sinopharm Chemical Reagent Co., Ltd and capsaicin (purity > 95%, batch number: P12D9S77366, Yuan Ye Biology Co., Ltd.).

## 2.3 Experimental grouping

The experimental animals were numbered according to body weight in ascending order. In the initial grouping, 10 animals were randomly selected as the blank control group(Group A) by using the random number table method, and the remaining 50 animals were treated by unified modeling. After successful modeling, the animals were randomly divided into 5 groups: Group B was the blank control group, Group B was the CVA model group (hereinafter referred to as the model group), Group C was the Montelukast sodium group, Group D, E and F were the low-dose group, mid-dose group and high-dose group of MLAD, respectively.

## 2.4 Model replication

Based on previous studies and the exploration of the CVA model by the previous research group [13], we adopted the following modeling methods: intraperitoneal injection of immunosuppressant cyclophosphamide (30 mg/kg) on the 1st day, 1 mL of 2 mg OVA and 200 mg aluminum hydroxide suspension on the 3rd day, and intraperitoneal injection of 0.1 mg ovalbumin and 100 mg aluminum hydroxide mixture on the 22nd day to enhance the immune response. From the 23rd day, 1% OVA solution atomization was performed (the atomization rate was 3 mL/min for 20 seconds and gradually extended to 90s), once every other day, for 7 times. The normal control group was given the same dose of normal saline. Evaluation of the animal model: after modeling, it was mainly assessed by the behavioral performance, the key manifestations are nodding and shrugging, coughing without obvious wheezing after stimulation, cough times observed more than 5 times within 3 minutes, mental irritability or sluggishness and so on. If the above conditions are met, it is determined that the modeling is successful.

## 2.5 Administration method and dosage

The dosage for each group was calculated according to the conversion formula of body surface area between humans and guinea pigs. From the 23rd day of the experiment, 30 minutes before the stimulation with a nebulizer, each group was given intragastric administration intervention for 14 days. The blank control group and model group were fed with drinking water 1mL/100g/d once a day. The C group was administered with Montelukast sodium suspension 1mg/kg (0.1mg/mL, 1mL/100g/day) orally once a day. D group: oral administration of 0.74g/mL of MLAD solution, 1mL/100g/d, once a day. E group: oral administration of 1.48g/mL MLAD solution, 1mL/100g/d, once a day. F group: oral administration of 2.22g/mL MLAD solution, 1mL/100g/d, once a day. The three concentration groups of MLAD are equivalent to 1:2:3 of the adult dosage.

## 2.6 Observation index

### 2.6.1 The general condition of CVA Guinea pigs.
The appearance (coat color, hair quality, mental state, etc.), abnormal behavior (cough, sneezing, accelerated breathing, abdominal twitching, etc.), appetite, and body weight were the primary observation indices for each group. Body weight was used as quantitative statistical data.

### 2.6.2 Airway sensitivity of CVA Guinea pigs.
Except for the blank control group, all the other guinea pigs were placed in a confined space to inhale the $10^{-4}$mol/L capsaicin solution (capsaicin was added to Tween-80 solution, absolute ethanol, and normal saline to prepare a $10^{-4}$mol/L capsaicin solution, ready to mix and use [14]) for 60 seconds after being

subjected to a 1% atomization challenge on the 36th day. Thereafter, the atomizer was switched off and the box lid was lifted after 60 seconds. The guinea pigs were then retrieved from the box, and the number of coughs for each guinea pig in 3 minutes was recorded.

**2.6.3 Airway resistance of CVA Guinea pigs.** The animal pulmonary function test (PFT) system was used to observe the changes in airway resistance (RI). The guinea pigs of each group were anesthetized by an intraperitoneal injection of 20% urethane solution at 1g/kg. The volumetric box was later sealed. After determining the base value, the stimulating reagent and sequence were as follows: 0.5mL of normal saline and 0.5mL of methacholine with concentrations of 0.02, 0.04, and 0.08 mg/mL for excitation. After each injection, the airway resistance at each concentration level was recorded. The subsequent injection was performed after the total RI returned to normal levels.

**2.6.4 Detection of IL-2, IL-4, IL-12 and IL-13 levels in serum and BALF by ELISA.** After completing PFT, $CO_2$ was euthanized. Immediately, blood was drawn from the abdominal aorta and balf was extracted by alveolar perfusion from the right lung. serum and balf supernatant were extracted by centrifugation after standing at room temperature for 2h. Elisa assay for the detection of IL2, IL4, IL12 and IL13 levels in serum and balf, according to the instructions of the relevant kits.

**2.6.5 Lung histopathology scoring.** After collecting BALF, the right lung hilar was quickly ligated, and the middle and lower lobes of the left lung were cut and preserved in formalin (Solarbio) for fixation 48h. The fixed lung tissue was routinely paraffin-embedded, sectioned at 4μm, deparaffinized, subjected to various levels of ethanol to water washing, water-dried and hematoxylin-stained for 5min, rinsed with water and blow-dried and then differentiated with hydrochloric acid-ethanol for 30s. It was soaked with water for 15min and then blow-dried, and then placed in eosin solution for 2min, and then dehydrated, clarified, and sealed with neutral resin, and photographed using an orthogonal fluorescence microscope (Leica DM3000).The quantitative grading criteria of the pathological sections [15] are shown in Table 2.

**2.6.6 Detection of Delta1, Jagged1, Notch1, and NICD proteins in lung tissue by Western blot.** After the collection of BALF, followed by rapid ligation of the left hilar, the tissue of the left upper lobe of the lung was taken and preserved in liquid nitrogen, and then the proteins were separated by gel electrophoresis, and the electrophoretically separated bands were transferred to NC/PVDF membranes, and the membranes were treated with antibodies against Delta1, Jagged1, Notch1, and NICD proteins, and finally the proteins were detected, imaged, and analyzed as a result. The optical density values of the target bands were analyzed by Quantity One image analysis software.

## 2.7 Statistical methods

The SPSS 26.0 statistical analysis software was used for statistical analysis. The experimental data were all measurement data, expressed as mean±standard deviation, in line with normal distribution, homogenous variance. The one-way

**Table 2. Histopathological Scoring System for Evaluating Changes in Pulmonary Inflammation in Guinea Pigs with Antigen Challenge.**

| Histopatholog-ical score | Eosinophilic infiltration in blood vessels and bronchi | Edema | Epithelial cell injury |
|---|---|---|---|
| 0 | Normal | Normal | Normal |
| 1 | Small amount of cell infiltration, no obvious histo-pathological changes | Mild diffuse edema | Mild cell injury |
| 2 | Small to medium amount of cell infiltration, mild tissue damage | Moderate alveolar and bronchiolar edema | Mild cell injury |
| 3 | Moderate cell infiltration and mild tissue injury | Regional and focal edema | Moderate cell injury |
| 4 | Moderate to large amount of cell infiltration, obvi-ous tissue damage | Significant edema | Moderate cell injury |
| 5 | Massive cell infiltration, severe tissue damage | Pneumatosis edema | Epithelial metaplasia, mucinous cell hyperplasia |

ANOVA was selected, and LSD was used for the pairwise comparison method test. If it did not conform to a normal distribution, the Kruskal-Wallis test was used. $p < 0.05$ was regarded as a statistically significant difference.

## 3. Experimental results

### 3.1 General condition of guinea pigs in each group

Throughout the whole process, the guinea pigs in the blank control group were sensitive and alert, had moist and smooth fur, loud calls, stable vital signs, a normal diet, and normal weight gain (body weight was measured every 3 days). In the model group, steady breathing was observed before OVA atomization. The guinea pigs gradually experienced varying degrees of rapid breathing, abdominal convulsions, coughing, and sneezing during the atomization process. There was also occasional nodding, shrugging, and face wiping observed in the guinea pigs. In terms of mental state, it was manifested as restlessness or sluggishness, and some guinea pigs would chew on the hair of other guinea pigs. With the increase in OVA atomization, the aforementioned symptoms, were progressively aggravated. The symptoms of each intervention group were similar to those of the model group at the initial stage of OVA atomization. After drug intervention, the mental state symptoms of the MLAD groups improved compared to those in the model group and the MS group.

On the 7th day after modeling, one guinea pig in the high-dose MLAD group died by accident. No abnormal findings were found after the autopsy. During the OVA atomization challenge, due to the large individual differences and different sensitivities to ovalbumin, 7 guinea pigs (2 each in the model group and the high dose group, and 1 each in the other groups) died during the atomization process. It was found that the lungs of died guinea pigs were hyperemic and the rest were normal. Therefore, the deaths could be due to high bronchial spasms and respiratory distress.

In terms of diet and weight, the guinea pigs in the high-dose MLAD group gained the most weight. The fur of the model group was slightly loose, and there was no significant increase in body weight in the later stage of the experiment, and some of the guinea pigs even lost weight. The body weight of the guinea pigs in each group was significantly lower than that of the blank control group, and the weight of the model group was significantly lower than that of the blank control group ($p < 0.05$). The body weight of the guinea pigs in the high-dose MLDA group was significantly higher than that of the other groups. There was no statistically significant difference amongst the other groups ($p < 0.05$). See Table 3.

### 3.2 MLAD reduces airway sensitivity and airway resistance in cva guinea pigs

The number of the number of coughs showed normal distribution and the variance was homogenous. Using the one-way analysis of variance, the cough frequency among the six groups was not the same. Compared with the blank control group [(1.80 ± 0.58) times] (normal saline atomization stimulation), the number of coughs in the model group [(11.00 ± 1.48) times], MS group [(6.40 ± 1.29) times] and low-dose MLAD group [(4.40 ± 0.93) times], mid-dose MLAD group [(5.00 ± 1.38) times] and high-dose MLAD group [(5.00 ± 1.14) times] were significantly increased, and the difference was statistically significant ($p < 0.05$). Compared with the model group, the number of coughs in each MLAD dose group and the MS group

**Table 3. Changes in Body Weight (g) of Guinea Pigs in each group before and after modeling.**

| Group | n | Before OVA induction | n | Before execution |
|---|---|---|---|---|
| Blank control group | 10 | 262.66 ± 1.73 | 10 | 436.20 ± 37.44 |
| Model group | 10 | 262.54 ± 6.52 | 8 | 384.86 ± 20.25* |
| MS group | 10 | 267.71 ± 3.68 | 9 | 408.38 ± 21.01 |
| Low-dose Modified Liu-an decoction group | 10 | 257.00 ± 4.87 | 9 | 404.00 ± 12.82 |
| Mid-dose Modified Liu-an decoction group | 10 | 261.80 ± 3.50 | 9 | 408.71 ± 19.70 |
| High-dose Modified Liu-an decoction group | 10 | 264.28 ± 7.16 | 7 | 457.80 ± 9.47# |

Note: Compared with the blank control group, *$p < 0.05$, Compared with the model group, # $p < 0.05$, Compared with the MS group, △$p < 0.05$.

was significant, with a statistically significant difference ($p<0.05$). The number of coughs in the low-dose MLAD group was the lowest. Compared with the MS group, the cough frequency in each MLAD dose group was lower, but there was no significant difference between the groups ($p>0.05$). (Table 4)

Compared with the blank control group, there were significant differences in airway resistance in normal saline, MeCH 0.02 mg/mL, and MeCh 0.04 mg/mL dose of the other five groups ($p<0.05$). There was a statistically significant difference in airway resistance between the low and high-dose MLAD groups and the model group at the MeCh 0.02 mg/mL dose. The airway resistance of the blank control group and the low and mid-dose MLAD groups gradually elevated with the increased stimulant concentration. On the other hand, the airway resistance of the model group, MS group, and high-dose MLAD group peaked at the MeCh 0.04 mg/mL dose.(Table 5 and Figs 1 and 2)

### 3.3 MLAD regulates Th1/2 cytokine expression levels

The IL-2, IL-12, and IL-13 levels in the model group were significantly higher, while the IL-4 levels were lower than those in the normal group ($p<0.05$). The IL-12 levels in the MS group and the MLAD groups were lower than those in the model group. The IL-2 levels in the MS group and the low-dose MLAD groups were significantly different from the model group ($p<0.05$). The IL-4 levels in each drug intervention group were higher than that in the model group, but there was no significant difference between the low and medium-dose MLAD groups and the model group. There was no statistically significant difference in IL-13 levels between each group and the model group (Fig 3).

### 3.4 Comparison of IL-2, IL-12, IL-4 and IL-13 levels in BALF of guinea pigs

The IL-2 and IL-12 levels in the model group were lower, and the IL-4 and IL-13 levels in the model group were higher than those in the blank control group ($p<0.05$). Compared with the model group, the IL-2 and IL-12 levels in the MS group and

**Table 4. Number of coughs of Guinea Pigs in each group after modeling.**

| Group | n | Cough frequency | *P | #P | △P |
|---|---|---|---|---|---|
| Blank control group | 10 | 1.80±0.58 | – | 0 | 0.011 |
| Model group | 8 | 11.00±1.48 | 0 | – | 0.011 |
| MS group | 9 | 6.40±1.29 | 0.011 | 0.011 | – |
| Low-dose Modified Liu-an decoction group | 9 | 4.40±0.93 | 0.13 | 0.001 | 0.24 |
| Mid-dose Modified Liu-an decoction group | 9 | 5.00±1.38 | 0.066 | 0.001 | 0.407 |
| High-dose Modified Liu-an decoction group | 7 | 5.00±1.14 | 0.066 | 0.001 | 0.407 |

Note: Compared with the normal group, *$p<0.05$, Compared with the model group, # $p<0.05$, Compared with the MS group, △$p<0.05$.

**Table 5. Comparison of the Area under the Total Airway Resistance Curve (RL-area) under Different Doses of Methacholine Challenge Test (cmH$_2$O/mL).**

| Group | Saline injection before stimulation | Acetylcholine content (mg/mL) | | |
|---|---|---|---|---|
| | | 0.02 | 0.04 | 0.08 |
| Blank control group | 48.6±7.46#△ | 72.54±5.69#△ | 90.35±12.11#△ | 110.99±19.48#△ |
| Model group | 171.34±11.69* | 206.02±11.29* | 299.14±23.55* | 246.92±76.05 |
| MS group | 174.86±50.93* | 188.19±54.98* | 240.18±66.41* | 138.03±35.91 |
| Low-dose Modified Liu-an decoction group | 113.61±23.02# | 122.13±30.77*# | 238.14±67.55* | 240.90±85.45 |
| Mid-dose Modified Liu-an decoction group | 162.02±21.31* | 175.64±22.03* | 224.37±42.90* | 328.87±112.60*△ |
| High-dose Modified Liu-an decoction group | 89.75±7.82#△ | 112.11±15.17#△ | 131.58±4.69#△ | 171.2±27.41 |

Note: Compared with the normal group, *$p<0.05$, Compared with the model group, # $p<0.05$, Compared with the MS group, △$p<0.05$.

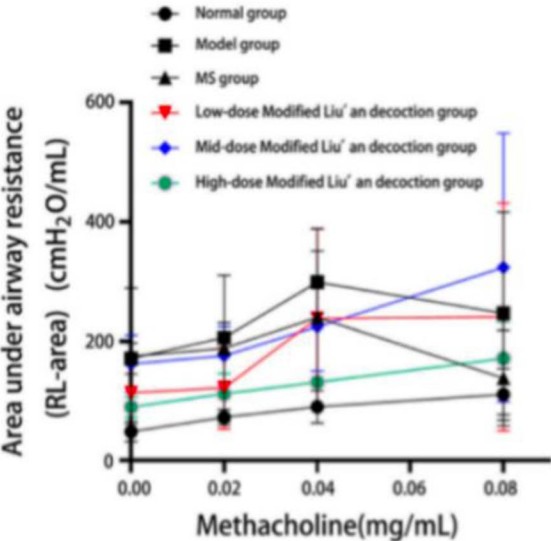

**Fig 1. Changes in airway resistance after excitation with different concentrations of acetylmethacholine.**

the various MLAD dose groups were higher, whereas the IL-4 and IL-13 levels were lower. Among them, the various cytokine levels in the MS group were significantly different from those in the model group ($p<0.05$). The IL-12 levels in each MLAD dose group were significantly higher than that of the model group ($p<0.05$). Among them, the IL-2 and IL-4 levels were significantly different between the high-dose MLAD group and the model group ($p<0.01$). There were significant differences between the MS group, the middle-dose MLAD group and the model group ($p<0.05$). The IL-13 levels decreased after treatment, with the low-dose MLAD group experiencing a significant decrease compared with the model group ($p<0.05$). However, there was no statistically significant difference between the middle-high-dose MLAD group and the model group. (Fig 4)

### 3.5 MLAD attenuates airway structural damage in cva guinea pig lung tissue

Regarding the histopathologic inflammation grading scores of guinea pig lungs in each group, the difference between the blank control group($2.4\pm0.25$) and the other 5 groups was statistically significant, and the difference between the low-dose MLAD group($6.3\pm0.44$) and the model group($8.1\pm0.43$) was statistically significant ($p<0.05$), there was no statistically significant difference in the middle-dose MLAD group($7.6\pm0.91$), MS group($7.5\pm0.50$) and the high-dose MLAD group ($7.1\pm0.91$). As shown in Figs 5 and 6 Histopathologic inflammation grading scores of guinea pig lungs in each group, the airway structure of the guinea pigs in the blank control group retained its normal integrity, with a clear and complete alveolar structure, without any evident inflammatory cell infiltration observed. The rest of the groups showed varying degrees of alveolar collapse, septal widening, partial alveolar dilatation, low columnar bronchial mucosal epithelium, an irregular proliferation of some airway mucosal epithelial cells, significant reduction of folds, as well as severe airway epithelial cell necrosis and shedding. There was also massive infiltration of monocytes, plasma cells, and eosinophils in the submucosal and alveolar septa, and hyperplasia of the airway smooth muscle. Furthermore, there was loosening, edema, and fibrosis around the bronchi and blood vessels.

### 3.6 MLAD inhibits NOTCH pathway expression

Compared with the normal group, the expression of the Notch ligand protein Delta1 in the remaining 5 groups was significantly decreased, while the expression of ligand Jagged1, receptor Notch1, and nuclear-localized Notch protein fragment

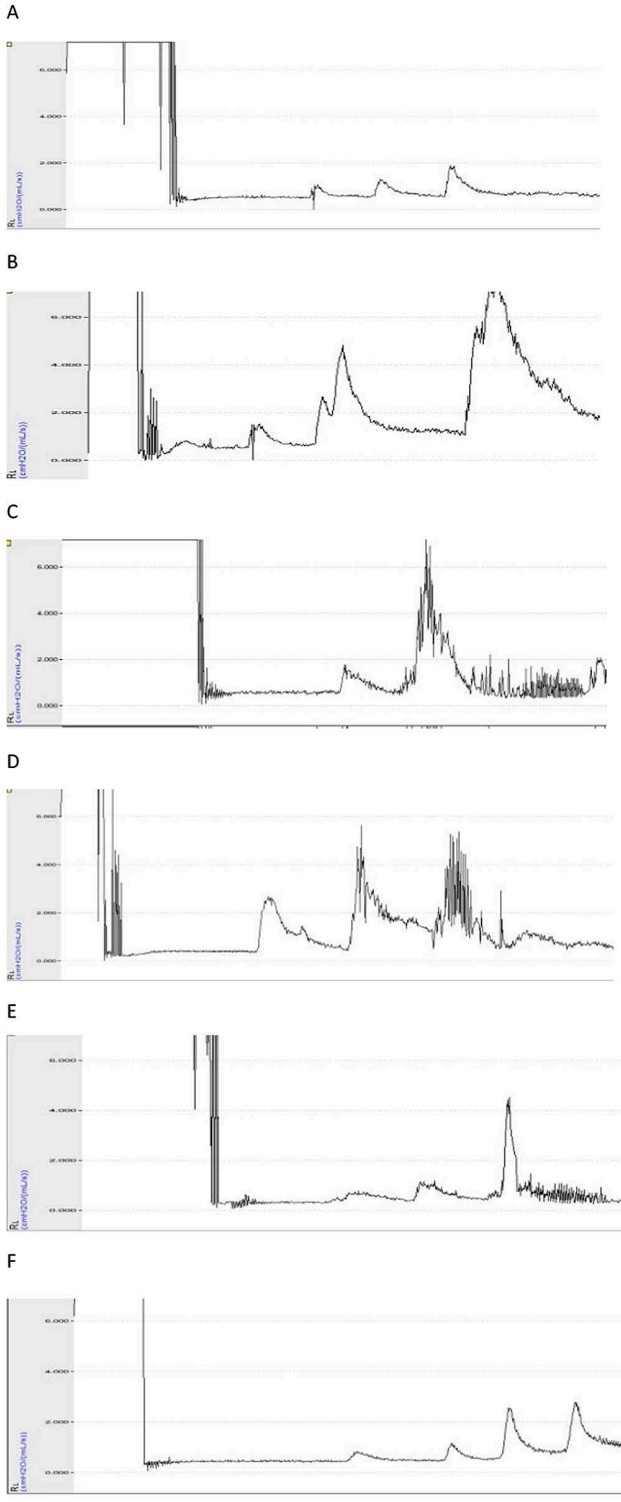

A:Blank control group B:Model group C:MS group D:Low-dose Modified Liu-an decoction group
E:Mid-dose Modified Liu-an decoction group F:High-dose Modified Liu-an decoction group

**Fig 2. Airway resistance curve during airway excitation in guinea pigsNote.**

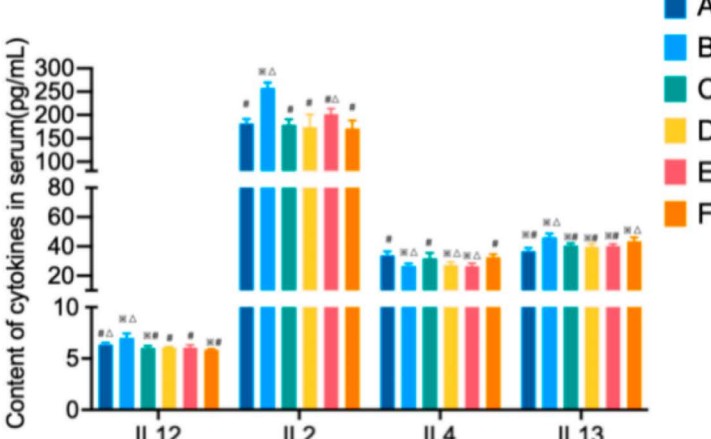

A:Blank control group B:Model group C:MS group D:Low-dose Modified Liu-an decoction group E:Mid-dose

Modified Liu-an decoction group F:High-dose Modified Liu-an decoction group

Note: Compared with the normal group, *P<0.05, Compared with the model group, #P<0.05, Compared with the

MS group, △P<0.05.

**Fig 3. Concentration of cytokines in the serum of guinea pigs in each group.**

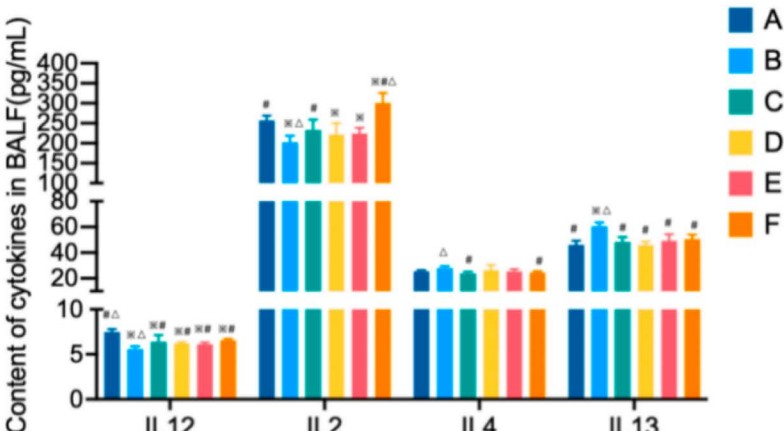

A:Blank control group B:Model group C:MS group D:Low-dose Modified Liu-an decoction group E:Mid-dose

Modified Liu-an decoction group F:High-dose Modified Liu-an decoction group

Note: Compared with the normal group, *P<0.05, Compared with the model group, #P<0.05, Compared with the

MS group, △P<0.05.

**Fig 4. Cytokine concentration in BALF of guinea pigs in each group.**

NICD was significantly increased ($p<0.05$). Compared with the model group, the expression of Jagged1, Notch1, and NICD proteins in the guinea pigs treated with MS and low-dose MLAD group decreased, while the expression of Delta1 protein increased ($p<0.05$). However, there was no statistically significant difference between the 2 groups, and there was

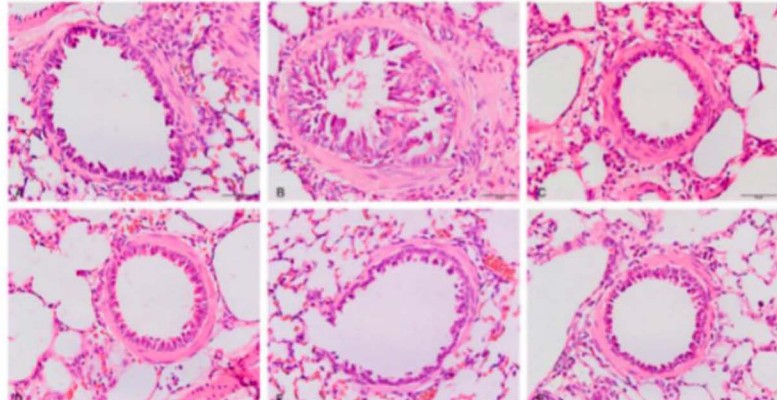

A:Blank control group B:Model group C:MS group D:Low-dose Modified Liu-an decoction group E:Mid-dose

Modified Liu-an decoction group F:High-dose Modified Liu-an decoction group

**Fig 5. HE staining results of lung tissue of guinea pigs in each group (×200,scale＝50μm).**

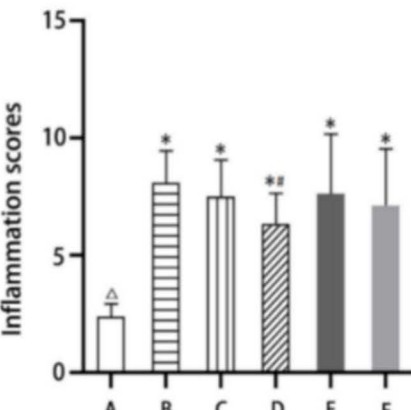

A:Blank control group B:Model group C:MS group D:Low-dose Modified Liu-an decoction group E:Mid-dose

Modified Liu-an decoction group F:High-dose Modified Liu-an decoction group

Note: Compared with the normal group, *$P<0.05$, Compared with the model group, #$P<0.05$, Compared with the

MS group, △$P<0.05$.

**Fig 6. Histopathologic inflammation grading scores of guinea pig lungs in each group.**

no statistically significant difference in protein levels between the middle-dose and high-dose MLAD group and the model group. (Figs 7 and 8)

## 4 Discussion

It is widely recognized that the pathophysiological process of CVA is not fundamentally different from that of typical asthma, and it is a subtype of bronchial asthma, with the same pathogenesis, which mainly includes airway immune-inflammatory mechanisms and neuromodulatory mechanisms and their interactions [16,17]. In this study, we investigated

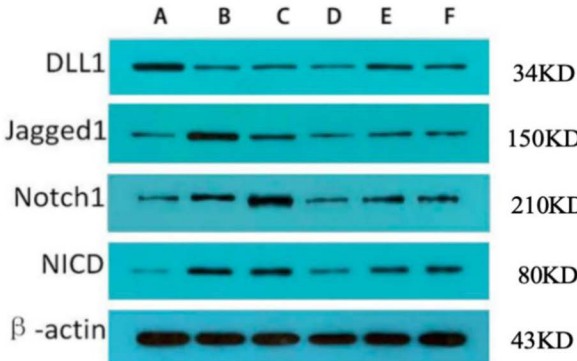

A:Blank control group B:Model group C:MS group D:Low-dose Modified Liu-an decoction group E:Mid-dose

Modified Liu-an decoction group F:High-dose Modified Liu-an decoction group

**Fig 7. Protein expression of Delta1, Jagged1, Notch1, and NICD in lung tissues by Western blot assay.**

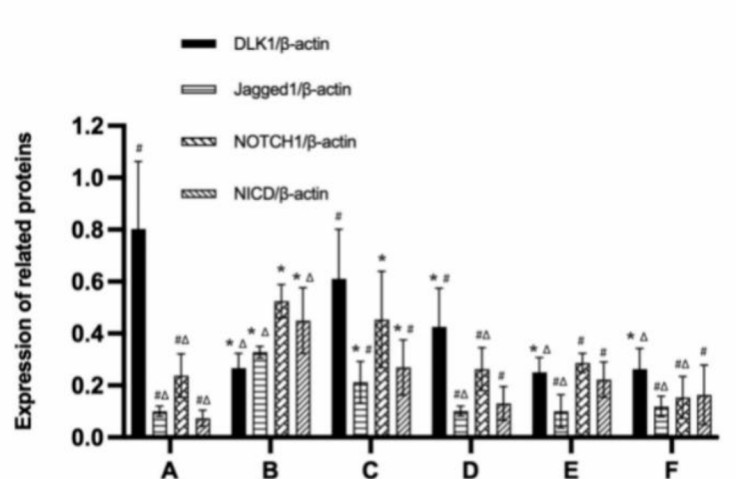

A:Blank control group B:Model group C:MS group D:Low-dose Modified Liu-an decoction group E:Mid-dose

Modified Liu-an decoction group F:High-dose Modified Liu-an decoction group

Note: Compared with the normal group, *P<0.05, Compared with the model group, #P<0.05, Compared with the

MS group, $^{\triangle}$P<0.05.

**Fig 8. Semi-quantitative analysis of Delta1, Jagged1, Notch1, and NICD protein levels in lung tissues.**

the efficacy of the traditional Chinese medicine Modified Liu-an Decoction in the treatment of CVA from the perspective of airway immune-inflammatory mechanisms, including airway inflammatory response, airway hyperresponsiveness, airway remodeling,and further explored specific mechanisms of action from the perspective of airway immunoinflammation.

## 4.1 Airway inflammatory

The inflammatory cells involved in the pathogenesis of CVA mainly include T lymphocytes, eosinophils (EOS), and mast cells. T lymphocytes play an essential role in the pathogenesis of CVA. By affecting the differentiation of Th1/Th2 cells,

the release of Th1 cytokines decreases and Th2 cytokines increases, resulting in chronic airway inflammation [18,19]. The decrease in the Th1/Th2 cell ratio is the initiation and maintenance factor of airway inflammation and airway hyperresponsiveness, and Th2 dominance is the central mechanism for the formation and progression of CVA [20]. Some studies have shown that children with CVA have higher levels of IL-4 and INF-γ/IL-4, and there is an imbalance of Th1/Th2 cells, which manifests as Th2 cell hyperactivity [21]. Therefore, by regulating the balance of Th1/Th2 cells and affecting the release of relevant cytokines, chronic airway inflammation in CVA children can be alleviated. The lung tissue contains the most Notch ligands and mRNA receptors. The Notch signaling pathway plays a significant role in lung tissue conduction, especially on the differentiation of Th1 and Th2 cells in T lymphocytes in the lung tissue. The Notch ligand Jagged is a dendritic-derived signal that plays a role in the differentiation of IL-4-independent Th2 cells, while the Notch ligand Delta is also a dendritic-derived signal that promotes the differentiation of Th1 cells [22]. Therefore, the Notch signaling pathway regulates T lymphocytes in the lung tissue through different ligand-receptor binding, so that the Th1/Th2 cells are in a state of relatively balanced state to keep the body healthy.

Among the observed indices in this study, IL-2 is mainly a 15kD glycoprotein secreted by Th1 cells, stimulated by IL antigens or mitogens, and induced by IL-1, which can stimulate the expression of MHC class II antigens in T cells and produce cytokines such as IFN-γ and TNF-α, which can promote the differentiation of Th0 cells into MHC and selectively inhibit the secretion of Th2 cytokines [23,24]. IL-12 appears earlier in the process and it can promote the differentiation of CD4+T cells into Th1 cells, and promote Th1 cells to secrete IL-2 and IFN-γ, thereby regulating the ratio of Th1 and Th2 to develop into Th1 [25,26]. Xu Hui et al [27] found that IL-12 could reduce the expression of IL-4 mRNA in the lung tissue of asthmatic mice, thereby reducing the chronic inflammatory infiltration of the airway. IL-4 is mainly secreted by Th2 cells with a variety of biological promoters, with the induction of the IgE-mediated immune response being the most important mechanism [28]. Matsukura et al [29] showed that IL-4 activates the inflammatory cells in the respiratory tract by upregulating the expression of eotaxin-1 and eotaxin-3 mRNA, which increases the number of acute attacks in asthmatic patients. IL-13 is produced by activated airway hyperresponsiveness (AHR)-related cells including Th2 cells, which can thicken the airway wall by promoting goblet cell proliferation [30] and mucus secretion [31]. When AHR-inducing factors are produced, IL-13 increases abnormally, which not only promotes the synthesis of IgE but also down-regulates the secretion of cytokines produced by Th1, such as IL-12 and IFN-γ, thus leading to the occurrence of disease [32,33]. IL-2 and IL-4 are key cytokines in the differentiation of Th0 cells into Th1 cells and Th2 cells. When the specific antigen-presenting cells are sensitized, they can differentiate into Th0A and Th0B. Th0A can secrete IL-2 and IL-4, while Th0B can secrete IL-2, IL-4, and IFN-γ. Under the action of high levels of IL-4, most Th0A and part of Th0B can differentiate into Th2. If the effect of IL-4 is blocked, Th0A will lose the ability to differentiate into Th2 and can express IFN-γ and differentiate into Th0B, and Th0B will further differentiate into Th1 [34]. IL-2 and IL-12 are also called NK cells and T cell growth factors respectively. The phenotypic difference between their inducing lymphocyte growth is that the main function of IL-2 is to promote the growth of T cells. On the one hand, it can stimulate T cells to produce cytokines such as IFN-γ and TNF-α after binding to the receptor and exerting its effector function. On the other hand, the IL-2 levels in the peripheral blood or plasma can also affect the balance shift of Th1/Th2 [35]. IL-12 mainly promotes the expression of NK cells and also maintains the growth of B cells. IL-12 can stimulate the production or apoptosis of various cells, promote the differentiation of Th0 into Th1, and produce a variety of cytokines such as IFN-γ, IL-2, GM-CSF, etc.[36]. IL-12 can reduce the number of eosinophils and the IgE levels in the lung tissue or blood by inducing eosinophil apoptosis and controlling the activation of mast cells [37]. IL-4 is the IgE regulator with the highest biological effect known to date, and IL-12 inhibits the activation of mast cells by reducing the production of IL-4, thereby reducing chronic airway inflammation [38]. IL-4 and IL-13 are Th2 cytokines that are closely linked in function and gene locus and have multiple overlapping functions. These functions play an important role in allergic airway inflammation and airway hyperresponsiveness. Their functional overlap is due to the α-chain of IL-4, which forms an important functional signaling component in the IL-4 and IL-13 receptors. The role of IL-4 and IL-13 in asthma is different. IL-4 may use its α- chain in the early and later stages when B cells are

isotype-switched to secrete IgE, while IL-13 uses this receptor to promote the production and development of airway hyperresponsiveness and inflammation in the late stage of the allergic reaction. In experimental studies, it was found that the IL-4 and IL-13 functions were positively correlated at the molecular level and in asthma [39–41]. The results of this study suggest that MLAD can reduce chronic airway inflammation by up-regulating the IL-2 and IL-12 levels in BALF, down-regulating the IL-4 and IL-13 levels, and regulating the dynamic balance of Th1/Th2.A number of domestic animal experimental studies have found [42] that traditional Chinese medicine compound can promote the expression of signal transducer and activator of ranscription4 (STAT4) in lung tissues by up-regulating the expression of helper T cell (Th)1 cytokine IL-12 and down-regulating the secretion of Th2 cells, thus controlling the development of the disease. of ranscription4 (Stat4) in lung tissues, reducing airway inflammation and thus controlling the development of the disease. The results of the study suggest that MLAD can regulate the dynamic balance of Th1/Th2 by up-regulating the content of IL-2 and IL-12 and down-regulating the content of IL-4 and IL-13 in BALF, thus reducing the chronic inflammation of airways, which is similar to the results of the above study.

The Notch signaling pathway plays an important role in lung tissue conduction, especially the differentiation of Th1 and Th2 cells in T lymphocytes in the lung tissue. The overexpression of the Notch1 and Jagged1 proteins can promote the expression of Th2 cell-specific transcriptional factor GATA3 and pro-inflammatory factor IL-4, thereby intensifying the polarization of Th2 cells in the body [43,44]. Experimental studies [45] have suggested that the expression of Jagged1, Notch1 proteins and their downstream transcriptional target Hes1 protein were significantly increased in OVA-induced chronic bronchial asthma mice. Therefore, the Notch signaling pathway regulates T lymphocytes in the lung tissue through different ligand-receptor binding, so that the Th1/Th2 cells are in a state of relatively balanced state to keep the body healthy. From the results of the Notch ligand/receptor protein levels in the lung tissue of the guinea pigs in each group, it was found that both MS and MLAD could increase the expression of Delta1 protein while inhibiting the expression of Jagged1, Notch1, and NICD proteins. The effect of the low-dose MLAD group was similar to that of the MS group. However, no salvage experiments were performed in this experiment, the specific downstream regulatory mechanisms need to be explored in further studies.

## 4.2 Airway hyperresponsiveness

In the results of this experiment, compared with the blank control group, the number of coughs of the guinea pigs in the other groups increased significantly after the last OVA protein atomization stimulation. The number of coughs in all groups was more than 5 times, suggesting that the model was successfully established. The number of coughs in the low-dose group was the least. The response of the guinea pigs to most protein antigens is a delayed hypersensitivity reaction mediated by specific sensitized effector T cells, which can be passively transferred by CD4＋T cells [46]. During the course of the experiment, it was also found that the cough was not very obvious on the day of atomization, and the guinea pigs in each group experienced more frequent coughing after drug administration the next day. After the intervention, the airway sensitivity of the guinea pigs in each group was stimulated with capsaicin atomization. The airway sensitivity of the guinea pigs in each treatment group was lower than that of the model group, suggesting that the intervention treatment was effective. The MLAD and MS sodium groups have similar efficacy in reducing airway sensitivity in CVA guinea pigs. In the pulmonary function challenge test, the airway resistance of the model group and MS group both peaked at the dose of MeCh0.04mg/m, considering that the guinea pig airway spasm has reached the limit at this concentration, and the airway spams have reached the limit when the concentration was increased. The smooth muscle of the airway failed to respond to increased concentrations of re-stimulation. However, the airway resistance of the normal group and each MLAD dose group gradually increased with the increasing concentration of the stimulant, suggesting that the airway hyperresponsiveness of the guinea pigs decreased after treatment with MLAD. The resistance value of the high-dose MLAD group was higher than that of the blank control group, but the trend was very consistent. It is considered that MLAD may increase the airway sensitivity threshold of CVA guinea pigs, thereby controlling neurogenic inflammation and reducing the number of

coughs. Nerve-related airway factors can increase tracheal vascular permeability, promote airway glandular secretion, and stimulate neurogenic inflammation such as airway smooth muscle spasm and contraction, while neuropeptide antagonists can effectively treat cough and relieve cough caused by capsaicin stimulation [47,48]. MLAD may also play a similar role in the treatment of CVA, but its specific mechanism requires further theoretical and experimental research.

### 4.3 Airway remodeling

Long-term chronic non-specific inflammatory stimulation of the airway can lead to varying degrees of pathological changes such as airway mucosal epithelial injury, cilia exfoliation and inflammatory cell infiltration, airway mucosal edema and hypertrophy, smooth muscle spasm, and airway inflammatory secretion obstruction in the airway. In this study, except for the blank control group, all the other groups had varying degrees of the aforementioned pathological changes. After the treatment with montelukast and MLAD, the pathological inflammation scores of the lung tissue of the guinea pigs in each treatment group decreased, indicating that the treatment was effective, and the curative effect of the low-dose MLAD group was better than that of the MS group. The hyperresponsiveness of the airway can restrict the airflow to different degrees. In mild cases, the pathological changes are reversible. However, if left unchecked, it will lead to airway remodeling [49]. Through Masson staining of the lung tissues of the guinea pigs in each group, we could see a thin layer of collagen deposition around the trachea of the guinea pigs' lung tissue in the blank control group, and a thicker layer of collagen deposition around the trachea of the guinea pigs' lung tissue in the model group. The collagen area and collagen volume fraction were significantly increased in the model group compared with the blank control group and significantly reduced in each treatment group. According to the Chinese medicine theory of "strange diseases have more phlegm" and "all diseases are caused by phlegm", it is believed that the key to the pathogenesis of cough variant asthma lies in the internalization of phlegm, which is caused by the deficiency of the lungs, spleen and qi, and the external cause is caused by wind evils that attack the lungs, and the combination of internal and external causes leads to the disease. From a microscopic point of view, this invisible phlegm in the lungs coincides with chronic inflammation of the airways, inflammatory cell infiltration, release of inflammatory factors, congestion and edema of the airway mucosa, increased mucus secretion, mucus plugs, collagen deposition and other modern pathologies, and the addition of Liuan Decoction to the lungs can clear lungs and remove heat and strengthen the spleen to resolve phlegm, congestion, edema degree, reduce airway inflammation, and at the same time can reduce peri-tracheal collagen deposition, effectively improve airway remodeling, which is consistent with the results of the previous rat experiments [11].

## 5 Conclusion

In this study, we explored the effectiveness and possible mechanisms of MLAD in the treatment of CVA. First we found that MLAD reduced the number of coughs and airway resistance in CVA guinea pigs. Secondly, MLAD effectively regulated the levels of inflammatory factors IL12,IL2,IL4,IL13 in serum and balf, inhibited airway inflammation, and at the same time attenuated inflammatory infiltration in lung tissue. MLAD could increase the expression of Delta1 protein while inhibiting the expression of Jagged1, Notch1, and NICD proteins, the specific downstream regulatory mechanisms need to be explored in further studies.In summary, we verified that mlad prevented CVA by ameliorating airway inflammatory infiltration and airway remodeling induced by th1/2 differentiation imbalance, and its mechanism may be related to the inhibition of Notch pathway expression.

## Author contributions

**Data curation:** Fangwei XU, Panpan Li, Chen Lu, Yuhang Chen, Ye Zhang.

**Formal analysis:** Fangwei XU, Liqun Wu.

**Funding acquisition:** Liqun Wu.

**Writing – original draft:** Fangwei XU, Ningning Zhang.

**Writing – review & editing:** Fangwei XU, Jian Deng, Ningning Zhang, Kimberly XinTing LEOW, Liqun Wu.

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
