## [Decision Letter · Decision Letter 0]

12 Nov 2024

Dear Dr. XU,

Thank you for submitting your manuscript to PLOS ONE. After careful consideration, we feel that it has merit but does not fully meet PLOS ONE’s publication criteria as it currently stands. Therefore, we invite you to submit a revised version of the manuscript that addresses the points raised during the review process.

We look forward to receiving your revised manuscript.

Kind regards,

Misbahuddin Rafeeq

Academic Editor

PLOS ONE

Journal Requirements:

6. Please upload a copy of Figures 1-10, to which you refer in your text on your manuscript. If the figure is no longer to be included as part of the submission please remove all reference to it within the text.

Reviewers' comments:

Reviewer's Responses to Questions

**Comments to the Author**

1. Is the manuscript technically sound, and do the data support the conclusions?

Reviewer #1: Yes

Reviewer #2: Partly

2. Has the statistical analysis been performed appropriately and rigorously?

Reviewer #1: No

Reviewer #2: Yes

3. Have the authors made all data underlying the findings in their manuscript fully available?

Reviewer #1: Yes

Reviewer #2: Yes

4. Is the manuscript presented in an intelligible fashion and written in standard English?

Reviewer #1: Yes

Reviewer #2: Yes

Reviewer #1: Dear authors,

The statistical methods presented in a transparent way help readers understand how the data was analyzed, increasing the reproducibility and credibility of the study. Sensitivity analyzes are responsible for testing the robustness of the results of different assumptions or analytical choices in the study. Defining the limitations and biases of the study are important to clarify to the reader the possible restrictions present in the research design, collection or analysis of the data obtained, providing an adequate interpretation of the results, in addition to helping to understand the potential impact of these factors, promoting transparency to readers. The statistical analyzes of the research were well executed, but I suggest presenting improvements in methodological transparency and sensitivity analyses, which could further increase the robustness and credibility of the results obtained. Providing a detailed description of the statistical methods used would also be interesting. Furthermore, researchers could define the limitations surrounding the study and define the biases present in the research, if any.

Reviewer #2: I acknowledge the efforts of the authors in an attempt to provide a therapeutic solution for asthmatic conditions. While the research addresses a clinically relevant question with potentially significant implications for treatment planning, several areas are noted with flaws and clinical concerns. Some of these are summarized below;

MAJOR CONCERNS:

1. CLINICAL CONCERN: In this study, the use of numerous herbs (more than 10) appears somewhat arbitrary, suggesting a trial-and-error approach by combining multiple ingredients to achieve results. No identifiable compounds other than concocting too many herbs without analyzing their constituents. This method raises health concerns due to the potential toxicity of unknown constituents. Employing such a mixture without a thorough analysis of each component’s active and potentially toxic compounds is neither scientifically rigorous nor acceptable for ensuring safety. Therefore, it is highly important to provide a detailed analysis of the constituents of the herbs used in a study like this, especially for a few key reasons:

a) Pinpoint specific compounds responsible for therapeutic effects to identify the active agents understanding that this knowledge allows researchers to attribute biological activity (such as anti-inflammatory, bronchodilatory, or immune-modulatory effects) to particular compounds. Without this, attributing effects solely to the herbal mixture may limit the scientific rigor of the study and make it difficult to determine which components are beneficial.

b) Detect and manage potentially harmful compounds. Herbs can contain toxic compounds that may pose health risks, particularly if they are administered in higher doses or for prolonged periods. Detailed analysis helps identify any potential toxins, ensuring safer use in clinical or experimental settings.

c) Facilitate consistent replication of treatment effects in future studies. If the herb mixture (like MLAD in this case) is to be recommended for future therapeutic use, having a clear understanding of its components is essential for reproducibility. It ensures that other researchers or clinicians can recreate the mixture with similar composition and dosage, leading to consistent results and therapeutic efficacy.

d) Understand how individual components affect biological pathways. If one herb contains a compound that modulates the Notch signaling pathway (central to the study’s hypothesis), identifying it could enhance understanding of how the herbal mixture exerts its effects on Th1/Th2 function.

2. POOR METHODOLOGY:

a) Experimental model: Using 3-week-old guinea pigs in a study involving respiratory function, immune response, and airway inflammation seem unethical and thus should be approached with caution. At this age, guinea pigs are quite young, and their respiratory and immune systems are still maturing, which could influence the results and interpretation of the study.

I. At 3 weeks, guinea pigs are weaned but still undergoing significant physiological development, including in the lungs. Immature lung structure and function could lead to different responses to asthma or inflammation compared to fully mature animals.

II. The immune response in young animals may not fully mimic that in adults, especially in conditions involving Th1/Th2 balance and chronic inflammation. Young guinea pigs may have an underdeveloped adaptive immune response, which could affect how they respond to allergens and treatments.

III. For studies aimed at understanding asthma-like conditions in humans, using older guinea pigs (e.g., 6–8 weeks or older) would generally provide a more stable model, as their respiratory and immune systems would more closely recapitulating those of adult humans. Young animals might not fully replicate the chronic and complex immune dynamics of asthma, especially cough variant asthma (CVA).

IV. Young animals may experience more stress and potential adverse effects from experimental procedures, particularly if they involve repeated allergen exposure and invasive testing and may therefore violate ethical considerations and animal welfare.

3. Regarding the statement, "MLAD was composed of Pinellia Ternata 6g, Citrus maxima 6g, Poria Cocos 6g, Prunus Armeniaca 6g, Sinapis Alba 6g, Glycyrrhiza uralensis 3g, pumice 20g, Lepidium Apetalum 6g, Semen Trichosanthis 10g, Rhizoma Arisaema Cum Bile 4g, Raphanus Sativus 10g ", some concerns are raised;

I. The statement does not provide any rationale for the specific quantities of each ingredient. While traditional formulations may prescribe certain amounts, discussing why these specific doses are used could enhance understanding, particularly for readers unfamiliar with TCM.

II. The study failed to show where each of the MLAD composition was sourced; root. fruits, stem or so. Different parts can contain varying concentrations of active compounds and may have distinct therapeutic effects. Including this detail would enhance the study’s reproducibility and scientific rigor.

III. Though minor, while scientific names are provided, they are inconsistent in capitalization and formatting. Scientific names should always be italicized, with the genus capitalized and the species name in lowercase (e.g., Pinellia ternata).

4. Under “1.3 Experimental grouping”, the grouping method used in this study is flawed due to a lack of consideration for body weight balance across the groups. Ensuring that each group has a relatively similar mean body weight is essential for minimizing variability that could affect treatment response and overall outcomes. However, this crucial step was overlooked. In Table 3, both the administration of ovalbumin and the treatments appear to be based on body weight when the animals were euthanized for analysis, and for the endpoint outcomes. This oversight introduces a significant bias, as variations in body weight on group level could influence the animals' responses to both the sensitization and treatment, potentially skewing the results. This lack of initial weight-based grouping represents a fundamental methodological flaw that weakens the study’s reliability and validity.

5. The behavioral assessment criteria used in this study are flawed and inadequate. The criteria for assessing successful model induction—such as nodding, shrugging, and coughing—are subjective and may result in inconsistent evaluations. Implementing a validated scoring model for behavioral assessment, such as the Asthma Severity Score or Pulmonary Symptom Scoring System, would improve objectivity. Such models assess symptoms based on standardized metrics like cough frequency, respiration rate, and activity level, providing consistent criteria that reduce observer bias and enhance the reproducibility of results.

6. Cyclophosphamide, a powerful immunosuppressant, is administered on Day 1 without clarification of its specific role. Given that CVA is an immune-driven condition, the inclusion of an immunosuppressant may affect the immune response’s development in ways not typical of CVA, potentially influencing the model's relevance.

7. The cytokine analysis in this study is limited by its focus on Th2 responses, limited from Th1.

a) It omits critical data on Th1 cytokines, such as IFN-γ and other pro-inflammatory markers, which are crucial for understanding the balance between Th1 and Th2 pathways in asthma. The study would benefit from investigating how Th1 cytokines might be downregulated in an asthmatic context, along with potential pathway crosstalk between Notch signaling and Th1/Th2 regulation.

b) Additionally, the role of effector cells such as macrophages and Tcells (both) in this model remains unexplored. Effector cell activity and differentiation are key to understanding the immune response and would add depth to the study’s conclusions. A simple flow analysis could have improved the study rigor.

c) For robust conclusions on cytokine expression, it’s essential to account for possible confounding factors such as underlying health conditions or infections (e.g., intestinal or blood pathogens) that can influence cytokine levels. Cytokine regulation is heavily influenced by the JAK-STAT pathway; yet, the study lacks details on how, or if, JAK-STAT interacts with Notch signaling. Clarifying these pathway interactions would strengthen the study’s insights into the mechanisms at play in asthma.

8. The authors claimed that they stained Masson Trichrome for fibrosis analysis in the lungs, only to be seen on the section slide. In fact, I had expected IHC stain to establish the findings here but to no avail.

MINOR CONCERNS:

1. The “Conclusion” section is placed before the “Discussion.” This is unconventional; the authors should justify this arrangement or consider reordering the sections.

2. There are numerous capitalization errors throughout the manuscript that need to be addressed.

3. In the dosing description, there’s an error in consistency and clarity: “Low-dose MLAD decoction group: oral administration of 0.74g/mL MLAD solution, 1mL/100g/d, once a day. Mid-dose MLAD group: oral administration of 1.48g/mL MLAD solution, 1mL/100g/d, once a day. Low-dose MLAD group: oral administration of 2.22g/mL MLAD solution, 1mL/100g/d, once a day. The three concentration groups of MLAD are equivalent to 1:2:3 of the adult dosage.” Something is wrong here. This should be clarified and corrected to accurately describe each dosage group.

4. The statement “From the day of stimulation, that is, from the 23rd day of the experiment, 30 minutes before the stimulation with a nebulizer, each group was given intragastric administration intervention for 14 days” lacks specificity. What was the purpose of this intervention?

5. The sentence, “The dosage for each group was calculated according to the conversion formula of body surface area between humans and guinea pigs,” is vague and should be clarified, perhaps with the formula provided for transparency.

6. Avoid starting sentences with numerals, as in “60 healthy 3-week-old pure male guinea pigs...” Instead, write, “Sixty (60) healthy 3-week-old male guinea pigs...” This applies throughout the manuscript.

7. Some sentences are overly lengthy and difficult to follow, such as: “Through our previous animal experimental studies, we discovered that MLAD could reduce the levels of serum tumor necrosis factor-α (TNF-α) and interleukin-5 (IL-5) in CVA rats, reduce the number of eosinophils (EOS) in the bronchoalveolar lavage fluid (BALF), reduce airway inflammation and airway collagen deposition, thereby improving the pathological state of airway remodeling and reducing cough frequency.” This should be broken down for clarity.

8. The authors mention a previous clinical trial: “Our previous clinical trials have shown that treatment of CVA using Modified Liu-an Decoction (MLAD) and Montelukast sodium was effective in children with phlegm-heat and phlegm-dampness syndromes.” Specific details, including location and timing, should be provided to give context to this claim.

9. Many statements appear to be derived from the literature without proper citation. For instance, “The etiology and mechanism of CVA are complicated, and antibiotic treatment is ineffective,” lacks a reference.

10. The abbreviation “CVA” is used without being defined at first use. It should be spelled out initially for clarity.

11. There is no “Introduction” section, which is unusual for a scientific manuscript.

12. The introduction should provide a rationale, emphasizing how this new agent could potentially offer advantages over current treatments and why it is important to pursue this line of research.

13. The poor design and readability of figures and graphs diminish the accessibility and interpretability of the data. Enhanced visuals with clear labeling would allow readers to better understand the results.

In summary, the study would benefit from a stronger structural foundation, improved clarity and consistency, more rigorous methodology, comprehensive data analysis, detailed justification for MLAD's use, and enhanced figures. Addressing these points would lead to a more reliable and scientifically robust study.

**Do you want your identity to be public for this peer review?** For information about this choice, including consent withdrawal, please see our Privacy Policy

Reviewer #1: No

Reviewer #2: **Yes: ** Kamoru Adedokun

---

## [Author Response · Author response to Decision Letter 1]

27 Dec 2024

The editors may not be particularly familiar with TCM medications; the medications used in this study have been clinically tested over a long period of time and there are clinical trials to prove their safety�HU Jiao Jiao. Study on the effect of slowing croup six-an decoction on small airway function in children with asthma in remission [D]. Beijing University of Traditional Chinese Medicine,2019.�. The guinea pigs have already entered puberty at 13 weeks, so we chose 3-week-old guinea pigs for the pediatric study, including the effects of the drug on their growth and development.

---

## [Decision Letter · Decision Letter 1]

24 Mar 2025

Dear Dr. XU,

Thank you for submitting your manuscript to PLOS ONE. After careful consideration, we feel that it has merit but does not fully meet PLOS ONE’s publication criteria as it currently stands. Therefore, we invite you to submit a revised version of the manuscript that addresses the points raised during the review process.

We look forward to receiving your revised manuscript.

Kind regards,

Misbahuddin Rafeeq

Academic Editor

PLOS ONE

Journal Requirements:

Reviewers' comments:

Reviewer's Responses to Questions

**Comments to the Author**

Reviewer #3: (No Response)

Reviewer #4: (No Response)

2. Is the manuscript technically sound, and do the data support the conclusions?

Reviewer #3: Yes

Reviewer #4: Yes

3. Has the statistical analysis been performed appropriately and rigorously?

Reviewer #3: Yes

Reviewer #4: Yes

4. Have the authors made all data underlying the findings in their manuscript fully available?

Reviewer #3: No

Reviewer #4: No

5. Is the manuscript presented in an intelligible fashion and written in standard English?

Reviewer #3: Yes

Reviewer #4: Yes

Reviewer #3: The authors have done great work, but there are still some questions and suggestions for them.

Q1: There are some format errors. For example, in the first line of the second paragraph of 2.1 General condition of guinea pigs in each group, the 7th should be 7th ("th" superscripted). The spacing before and after punctuation marks (such as “,”, “<” and “±”) is inconsistent. Also, the "n"s indicating sample size and the "P"s indicating significance should be italicized. The capitalization of 3.3 is not consistent with that of 3.1 and 3.2. I hope the authors can improve the format of this manuscript.

Q2: Figure 1, 3, 4, 6, 7, 8 should each include a title, not just a legend.

Q3: In Figure 7, the protein weights should be written as kDa and the font should be standardized (I suggest that they be standardized to Arial).

Q4: I suggest that the bars in Figure 8 should be grouped by proteins, rather than experimental groups. This could help with the clarity of the results. Also, I think the * and # showing significance are misplaced, as they are not positioned directly above the bars.

Q5: Is it possible to add some indicators that can directly demonstrate Th1/Th2 conversion, such as ① flow cytometry or ② the detection of Th1 marker T-box transcription factor (T-bet) and Th2 marker GATA-binding protein 3 (GATA-3) or ③ Th1-associated proinflammatory cytokines interferon (IFN)-γ and tumor necrosis factor (TNF)-α by CD4+ T cells?

Q6: I suggest that the original Western Blot data should be provided in supplementary materials.

Q7: There are some Chinese characters in the 3 Discussion part of Revised Manuscript with Track Changes.

Reviewer #4: Comments to the Authors

First, I would like to declare that I have reviewed the Revised Manuscript with Track Changes. The manuscript by Fangwei Xu et al., titled "Study on the Effect of Modified Liu-an Decoction on Th1/Th2 Function in Guinea Pigs with Cough Variant Asthma through the Notch Signal Pathway", concludes that the potential mechanism of Modified Liu-an Decoction (MLAD) in treating CVA involves regulating the dynamic balance of Th1/Th2 by modulating the expression of Notch pathway-related proteins, thereby alleviating chronic airway inflammation and improving airway remodeling. I find this study highly significant, but there are issues with its design and manuscript presentation that I hope the authors can address and improve.

Comments:

1.The study design includes six groups labeled A-F, where Group A is the blank control group, Group B is the model group, and the remaining groups (C-F) can be referred to as model drug groups (MS group and MLAD groups). I believe the primary role of Group A is to validate the successful modeling of Groups B-F. The authors also indicate that the animal model is mainly evaluated based on behavioral performance, with the effective observation indicators being 1.6.1-1.6.2. When analyzing the experimental data, considering the influence of a single factor, comparisons between the blank control group and the model group, as well as between the model group and the model drug groups, are statistically meaningful. However, in the description of the experimental results, there are comparisons between the blank control group data and the model drug group data, such as in 2.1 regarding diet and weight, 2.2 regarding airway sensitivity, 2.3 regarding airway hyperresponsiveness, and 2.6 regarding lung tissue of guinea pigs in each group.

2.The manuscript exhibits a lack of precision in its presentation. For instance, in the Introduction, the phrase "[7]" is used; in Section 1.2, the notation "AI(OH)3" appears; in Section 1.3, the text states "divided into 5 groups"; in Section 1.5, there is a duplication of "E group" and an omission of "F group"; in Section 1.7, the term "mean±standard" is employed; and in Sections 2.4 and 2.7, the expression "the normal group" is utilized. These inaccuracies and inconsistencies require correction.

3.Certain paragraphs are overly lengthy and would benefit from appropriate segmentation. For example, in the Discussion, the second paragraph regarding Airway Inflammatory, as well as the sections on Airway Hyperresponsiveness and airway remodeling, could be restructured for improved clarity and readability.

4.In the Introduction, the authors state, "Through our previous animal experimental studies, we discovered that MLAD could reduce the levels of serum tumor necrosis factor-α (TNF-α) and interleukin-5 (IL-5) in CVA rats, reduce the number of eosinophils (EOS) in the bronchoalveolar lavage fluid (BALF), reduce airway inflammation and airway collagen deposition, thereby improving the pathological state of airway remodeling and reducing cough frequency." However, in the Results section of the Abstract, the authors mention, "Compared with the model group, MLAD reduced the inflammatory infiltration and collagen deposition in the lungs of CVA guinea pigs (P<0.05)." This raises the question of whether the same data has been reported in multiple publications.

5.Due to the current font used, certain details in the figures are difficult to discern, particularly in Fig. 2, Fig. 3, and Fig. 4. It is recommended to enhance the clarity and resolution of these figures to ensure accurate interpretation of the data.

**Do you want your identity to be public for this peer review?** For information about this choice, including consent withdrawal, please see our Privacy Policy

Reviewer #3: No

Reviewer #4: No

---

## [Author Response · Author response to Decision Letter 2]

29 May 2025

Thanks for the valuable suggestions for changes, not required at this time

---

## [Editor Report · Decision Letter 2]

15 Jun 2025

Study on the Effect of Modified Liu-an Decoction on Th1/Th2 Function in Guinea Pigs with Cough Variant Asthma through the Notch Signal Pathway

PONE-D-24-12851R2

Dear Dr. Fangwei XU,

We’re pleased to inform you that your manuscript has been judged scientifically suitable for publication and will be formally accepted for publication once it meets all outstanding technical requirements.

Kind regards,

Misbahuddin Rafeeq

Academic Editor

PLOS ONE

---

## [Editor Report · Acceptance letter]

PONE-D-24-12851R2

PLOS ONE

Dear Dr. XU,

I'm pleased to inform you that your manuscript has been deemed suitable for publication in PLOS ONE. Congratulations! Your manuscript is now being handed over to our production team.

Kind regards,

on behalf of

Dr. Misbahuddin Rafeeq

Academic Editor

PLOS ONE